# Tumor Necrosis Factor α and Interleukin-1β Acutely Inhibit AgRP Neurons in the Arcuate Nucleus of the Hypothalamus

**DOI:** 10.3390/ijms21238928

**Published:** 2020-11-25

**Authors:** Fernanda M. Chaves, Naira S. Mansano, Renata Frazão, Jose Donato

**Affiliations:** 1Departamento de Fisiologia e Biofísica, Instituto de Ciencias Biomedicas, Universidade de Sao Paulo, Sao Paulo 05508-000, Brazil; fernandamchaves@gmail.com; 2Departamento de Anatomia, Instituto de Ciencias Biomedicas, Universidade de Sao Paulo, Sao Paulo 05508-900, Brazil; naira.mansano@gmail.com

**Keywords:** cytokines, inflammation, interleukin-6, obesity, POMC, sepsis

## Abstract

Obesity-associated low-grade inflammation favors weight gain, whereas systemic infection frequently leads to anorexia. Thus, inflammatory signals can either induce positive or negative energy balance. In this study, we used whole-cell patch-clamp to investigate the acute effects of three important proinflammatory cytokines, tumor necrosis factor α (TNF-α), interleukin-6, and interleukin-1β (IL-1β) on the membrane excitability of agouti-related peptide (AgRP)- or proopiomelanocortin (POMC)-producing neurons. We found that both TNF-α and IL-1β acutely inhibited the activity of 35–42% of AgRP-producing neurons, whereas very few POMC neurons were depolarized by TNF-α. Interleukin-6 induced no acute changes in the activity of AgRP or POMC neurons. Our findings indicate that the effect of TNF-α and IL-1β, especially on the activity of AgRP-producing neurons, may contribute to inflammation-induced anorexia observed during acute inflammatory conditions.

## 1. Introduction

The hypothalamus is a brain structure responsible for the regulation of numerous visceral functions, including the control of body temperature, autonomic nervous system, and feeding behavior, among others [1]. Particularly regarding the regulation of energy homeostasis, hypothalamic lesions can produce striking changes in hunger and body weight [2,3]. Several hypothalamic nuclei are involved in the central control of metabolism, even though a well-characterized population is composed of neurons located in the ventral aspects of the third ventricle. In rodents, this area is called the arcuate nucleus (ARH) and is composed of several neurochemically defined neuronal populations [4,5,6,7]. In the ventromedial aspects of the ARH, there is a great number of neurons co-expressing agouti-related peptide (AgRP) and neuropeptide Y (NPY) [8]. AgRP/NPY-producing neurons are powerful inducers of hunger, and their activation leads to weight gain [9,10]. A different group of neurons are spread over the lateral part of the ARH and they express proopiomelanocortin (POMC), which is a prohormone cleaved into different peptides, including α-melanocyte-stimulating hormone [11,12]. In contrast to AgRP/NPY-expressing cells, POMC neurons generally promote satiety [13,14].

An important aspect of ARH neurons is that they are very close to the median eminence, which is one of the circumventricular organs of the brain. Therefore, this region possesses a blood-brain barrier less selective than that in other brain areas [15]. Consequently, ARH neurons are exposed to circulating factors such as hormones, cytokines, nutrients, and toxins [15,16]. On the one hand, this characteristic allows ARH neurons to be an interface between what happens on the periphery to generate central responses [17]. However, ARH neurons are more vulnerable to insults or large variations in the levels of circulating factors [16,18]. In this regard, numerous pieces of evidence indicate the inflammatory status as a major factor that affects hypothalamic neurocircuits that control energy homeostasis, particularly ARH neurons [18,19,20,21]. Thus, both AgRP and POMC neurons are affected by cytokines involved in inflammatory responses [18,19,20,21].

Changes in hypothalamic inflammation are able to produce the opposite effects on energy homeostasis. It is well-known that obesity is characterized by a low-grade inflammation that affects several organs, including the adipose tissue, liver, gut, and hypothalamus [18,19,20,21]. In addition, obesogenic aspects in the environment, such as the consumption of saturated fatty acids [22,23] and alterations in the intestinal microbiota [24] are associated with systemic inflammation and altered energy metabolism. Therefore, obesity-associated low-grade inflammation is considered a factor that disturbs the ability of hypothalamic neurons to control energy homeostasis, predisposing individuals to metabolic imbalances [19,24]. Accordingly, high-fat diet intake induces acute increases in the hypothalamic expression of proinflammatory cytokines [20]. On the other hand, systemic infection or acute inflammatory conditions produce a sickness response in the central nervous system, frequently associated with fever, anorexia, anhedonia, and sleepiness [25]. For example, lipopolysaccharide (LPS)-treated mice exhibit reduced food intake [26]. In addition, LPS blunts the feeding response induced by AgRP neuronal activation [27]. Thus, it is clear that, depending on their magnitude, duration, and characteristics, hypothalamic inflammatory signals can either favor a positive or negative energy balance [25].

Although the consequences of the hypothalamic inflammation have been widely investigated, one aspect that has been less studied involves the effects of proinflammatory cytokines on the electrical activity of AgRP/NPY and POMC neurons. Scarlett et al. [28] showed that the proinflammatory interleukin-1β (IL-1β) increases the frequency of action potentials in ARH POMC neurons, which is expected to reduce hunger. However, whether other important proinflammatory cytokines, like tumor necrosis factor α (TNF-α) and interleukin-6 (IL-6), are also able to induce acute effects of the electrical activity of AgRP/NPY and POMC neurons is still unknown. Therefore, the objective of the present study was to investigate the acute effects induced by TNF-α, IL-6, and IL-1β on the resting membrane potential (RMP) and input resistance (IR) of AgRP/NPY and POMC neurons in the ARH. These findings can contribute to the understanding of how proinflammatory cytokines affect neural circuits that regulate food intake and metabolism.

## 2. Results

### 2.1. Characterization of Resting Biophysical Properties of AgRP Neurons

Although AgRP and POMC neurons are found close to each other in the ARH, they represent different cell populations [4,5,6]. Before evaluating the effects of cytokines in the activity of ARH neurons, using histological methods [29] we confirmed the exclusive expression of AgRP in the ventromedial part of the ARH using the AgRP-reporter mice (Figure 1A). In total, 50 AgRP neurons were evaluated from 24 mice. From these cells, 37 AgRP neurons (74%) were considered active because they were firing action potentials during the basal recording, whereas the remaining cells (13/50; 26%) were quiescent. The average RMP of active AgRP neurons was −52.6 ± 0.7 mV (range from −45 mV to −63 mV) and the mean resting IR was 1.8 ± 0.1 GΩ (ranging from 0.9 GΩ to 2.9 GΩ). The RMP of quiescent AgRP neurons was −52.5 ± 2.5 mV (range from −40 mV to −70 mV) and the mean resting IR was 1.0 ± 0.1 GΩ (ranging from 0.4 GΩ to 1.8 GΩ). 

### 2.2. TNF-α Acutely Inhibits the Activity of AgRP-Producing Neurons

The acute effect of the proinflammatory cytokine TNF-α was determined in brain slices of AgRP-reporter mice. For this purpose, 21 AgRP neurons were recorded from 15 mice. From the recorded cells, 17 of them were active, whereas four cells were quiescent. The results of active and quiescent cells are presented separately. Regarding the active cells, TNF-α induced a fast and significant hyperpolarization in six cells, representing 35% of the recorded AgRP neurons (Figure 1A). In this sense, TNF-α changed the RMP of responsive cells in −9.8 ± 0.6 mV (*p* < 0.0001, Figure 1A). In addition, IR was significantly reduced after TNF-α administration (−0.34 ± 0.11 GΩ, *p* = 0.0391, Figure 1A). It is worth mentioning that the TNF-α effect was not reversible during the washout period recorded (Figure 1A). In contrast, 11 AgRP neurons (65% of the cells) did not exhibit significant changes after TNF-α application to the bath (Figure 1B). All quiescent cells (*n* = 4) did not present significant alterations in their RMP and IR after TNF-α administration (Figure 1C). Thus, TNF-α produces an acute inhibitory effect on a subgroup of active AgRP neurons.

### 2.3. Activity of AgRP Neurons Is Not Acutely Affected by IL-6

The potential effects of IL-6 on the RMP and IR of AgRP neurons were also determined. The effects of an acute application of IL-6 were analyzed in 10 AgRP neurons from four mice. Half of the recorded neurons were active and the other half were quiescent. In contrast to the effects caused by TNF-α administration, we found that IL-6 did not significantly change the RMP and IR of either active (Figure 2A) or quiescent (Figure 2B) AgRP neurons.

### 2.4. AgRP Neurons Are Acutely Inhibited by IL-1β

To determine the acute effects of IL-1β, 19 AgRP neurons were recorded from seven mice. From the recorded cells, 15 of them were active, whereas four cells were quiescent. IL-1β induced a statistically significant hyperpolarization in eight cells, which represents 42% of the recorded AgRP neurons. In the responsive neurons, IL-1β induced a −6.2 ± 1.1 mV change in the RMP (*p* = 0.0007, Figure 3A). IR was also significantly reduced after IL-1β treatment (−0.24 ± 0.09 GΩ, *p* = 0.0442, Figure 3A). Similarly to TNF-α, the effect induced by IL-1β was not reversible during the recorded period (Figure 3A). Regarding nonresponsive cells, 11 AgRP neurons (58% of the cells) did not exhibit significant changes in the RMP and IR after IL-1β administration (Figure 3B). Since a previous study found that IL-6 and IL-1β act synergistically in the brain to induce sickness behavior [30], we also studied the effects of the co-administration of IL-6 in AgRP neurons in the initial responses induced by IL-1β. After the isolated infusion of IL-1β, IL-6 was co-administered in seven AgRP neurons (recorded from four mice). Similar to the previous results, IL-1β induced a hyperpolarization of the RMP in approximately 42% of AgRP neurons (3 out of 7 cells, change in RMP: −6.6 ± 2.3 mV, Figure 3C). The co-administration of IL-6 neither enhanced nor inhibited the response to IL-1β (*p* = 0.99, IL-1β versus IL-6 + IL-1β, Figure 3C). Taken together, IL-1β causes an acute inhibitory effect on the activity of ARH AgRP-expressing neurons.

### 2.5. Characterization of Resting Biophysical Properties of ARH POMC Neurons

POMC-expressing cells were visualized in the ARH (Figure 4A) and in the nucleus of the solitary tract (data not shown), as previously demonstrated [31,32]. Differently than AgRP neurons, POMC-expressing cells are spread in the ARH, especially in the lateral part (Figure 4A). In total, 35 POMC neurons in the ARH were evaluated from 22 POMC-reporter mice. From the recorded cells, 31 POMC neurons were firing action potentials during the baseline period (88%), whereas four cells were quiescent. The mean RMP of active POMC neurons was −52.1 ± 0.8 mV (range from −42 mV to −61 mV) and the average resting IR was 1.7 ± 0.1 GΩ (range from 0.5 GΩ to 3.3 GΩ). In quiescent POMC neurons, the RMP was −53.5 ± 4.2 mV (−42 mV to -62 mV) and the average resting IR was 1.0 ± 0.2 GΩ (0.7 GΩ to 1.5 GΩ).

### 2.6. TNF-α Causes Mild Effects on the Activity of POMC Neurons

To determine the potential effect of TNF-α on the biophysical properties of POMC neurons in the ARH, 21 cells from 13 POMC-reporter mice were analyzed. Four quiescent neurons were unwittingly analyzed, whereas 17 POMC cells were active (81%) at the basal period. Regarding the active cells, 15 POMC neurons did not exhibit significant alterations in the RMP (−0.3 ± 0.4 mV) or IR (+0.0 ± 0.1 GΩ) after TNF-α administration (Figure 4A). Of note, two POMC neurons from the same animal, but in different brain slices, exhibited a significant depolarization after TNF-α application and this effect was reversible during the washout period (Figure 4B). Due to the small number of POMC neurons that were responsive to TNF-α, no statistical analysis was performed. TNF-α did not produce significant changes in the RMP and IR of quiescent POMC neurons (Figure 4C).

### 2.7. IL-6 Causes No Acute Changes in the Activity of ARH POMC Neurons

The potential effects of an acute IL-6 administration were determined in 14 ARH POMC neurons from nine mice. In this case, all POMC neurons were firing action potentials during the basal recording period. Similar to the effects observed in AgRP neurons, we found that IL-6 did not acutely change the RMP (−0.1 ± 0.5 mV) or the IR (−0.1 ± 0.1 GΩ) of POMC neurons (Figure 5).

## 3. Discussion

Inflammatory signals in the central nervous system can produce paradoxical effects on food intake and body weight. During nutrient excess, inflammatory factors in the hypothalamus favor obesity, whereas, in systemic and/or acute infection, these signals cause anorexia and weight loss [25]. To improve our understanding of how proinflammatory cytokines affect neural circuits involved in the central regulation of metabolism, we determined whether the membrane excitability of ARH AgRP/NPY and POMC neurons is acutely affected by TNF-α, IL-6, and IL-1β. AgRP and POMC neurons were visualized by Cre-dependent reporter mice. Although developmental aspects may produce some false positive cells in these mouse models [33], the distribution of AgRP-reporter or POMC-reporter cells are in accordance with the known expression of these transcripts in the mouse brain [5,8,11,32]. Additionally, we have successfully used these mouse models in former histological and electrophysiological experiments [12,31,34,35].

Previous studies from our group employed whole-cell patch-clamp to determine possible changes in membrane excitability induced by different hormones/cytokines, including prolactin [36,37] and the growth hormone [34,38]. This method allows a precise determination whether the dominant hormonal action is stimulatory or inhibitory. However, a limitation of our experimental design is that we only evaluate acute cellular effects. In addition, the doses used are relatively high, even though they have been employed in previous in vitro studies [28,39,40]. In this sense, our experiments likely reproduce what happens during a strong, acute systemic inflammation, such as sepsis [25], rather than the chronic low-grade inflammation observed in obesity [19,20,24]. Acute systemic inflammation is frequently followed by a sickness response, which, among other symptoms, causes anorexia, weight loss, and cachexia. Importantly, this inflammation-induced anorexia is driven by the action of cytokines in the central nervous system [25]. In this sense, our findings indicate that the acute effects of TNF-α and IL1-β on AgRP-producing neurons likely contributes to the inflammation-induced anorexia since both cytokines inhibit the activity of 35–42% of AgRP neurons recorded. Of note, AgRP neurons induce hunger [6,9,10], so its inhibition is expected to decrease food intake and leads to weight loss. Another study also showed that either LPS or TNF-α blocks the activation of AgRP/NPY neurons by decreased glucose, which may alter glucose and energy homeostasis [41]. Additionally, our results highlight the importance of ARH AgRP-producing neurons as key targets of cytokines to produce the anorexic response for acute inflammatory signals. 

TNF-α may also enhance satiety by increasing the activity of a subgroup of POMC neurons since a previous study found that TNF-α induces a slight membrane depolarization in spontaneously active POMC neurons (3 out 10 cells analyzed) or in cells that received depolarizing current steps [42]. In our study, only two POMC neurons depolarized after TNF-α application and, interestingly, these cells were spontaneously active before TNF-α stimulus, which is in agreement with previous studies [42]. Taken together, the effect of TNF-α on the membrane excitability of ARH POMC neurons seems to be relatively lesser in comparison with the results observed in AgRP neurons. Nonetheless, the stimulatory effect of TNF-α on ARH POMC neurons is in accordance with in vivo data indicating that central TNF-α infusion inhibits food intake in rodents [43,44,45,46]. We did not evaluate the effects of IL-1β on the activity of POMC neurons because a previous study already showed that intracerebroventricular infusion of IL-1β induces Fos expression in ARH POMC neurons, and electrophysiological recordings demonstrated that IL-1β increased the frequency of action potentials and the release of α-MSH of ARH POMC neurons [28]. Thus, our observation that IL-1β causes an inhibitory effect in AgRP/NPY neurons is consistent with the well-known antagonistic role of POMC and AgRP/NPY neurons on feeding [14].

All TNF-α responsive cells were active (firing action potentials) before drug application, whereas no quiescent AgRP or POMC neuron was altered by this cytokine. Although the exclusive responsiveness of active cells may be coincidental, this response pattern likely has a biological significance. In this sense, TNF-α may act predominantly by blocking the feeding response induced by active AgRP-producing neurons, which are supposedly inducing the hunger feeling at that time. Another study also showed that LPS prevents the increase in food intake induced by AgRP neuronal activation [27]. 

TNF-α and IL1-β reduced the IR in AgRP-responsive neurons. IR is affected by the number of open ion channels in the cell membrane. The opening of ion channels decreases the membrane resistance. Thus, our findings suggest that TNF-α and IL1-β inhibit AgRP-producing neurons by increasing the number of ion channels opened. K^+^ or Cl^−^ channels are potential candidates to mediate the hyperpolarizing effect of TNF-α and IL1-β on AgRP neurons. However, since our extracellular solution contained 131.8 mM of Cl^−^ and the pipette solution contained 15 mM of Cl^−^, the equilibrium potential for Cl^−^ is expected to be ≈ −57 mV, which is close to the average RMP of responsive AgRP neurons (−52.5 ± 1.6 mV). On the other hand, the equilibrium potential for K^+^ is ≈ −101 mV. Thus, TNF-α and IL1-β likely altered potassium conductance, rather than chloride channels, to promote the hyperpolarizing effect observed in AgRP neurons.

IL-6 is a cytokine frequently released during inflammatory situations and the IL-6 receptor is expressed in the mouse ARH, including AgRP/NPY-producing and POMC-producing neurons [47]. However, the role of IL-6 in metabolism seems to be complex. Acute peripheral or central infusion of IL-6 does not affect food intake, even though IL-6 knockout mice develop late-onset obesity due to reduced energy expenditure [48]. IL-6 also does not inhibit food intake in chicks [49]. Thus, the absence of changes in the activity of AgRP and POMC neurons after IL-6 application is in accordance with the lack of IL-6 effects on feeding behavior. However, reduction in the biological activity of peripherally released IL-6 attenuates sickness behavior in rats [50]. Furthermore, intracerebroventricular co-infusion of IL-6 and IL-1β in doses that do not affect food intake when administered individually causes anorexia [30]. Therefore, IL-6 is involved in the sickness response even though its capacity to affect neuronal circuits that control food intake may depend on the combined effects of other pro-inflammatory cytokines, such as IL-1β. However, our findings indicate that the co-infusion of IL-6 did not affect the overall response induced by IL-1β on the activity of AgRP neurons. 

In conclusion, we demonstrated that TNF-α and IL-1β induce an acute inhibition in the electrical activity of ARH AgRP-expressing cells. This inhibitory effect helps to explain the anorexia caused by acute inflammatory conditions and points out AgRP neurons as key downstream targets of inflammatory signals to affect metabolism, even though a minor effect on POMC neurons may also exist [28,42]. Our electrophysiological findings are in accordance with in vivo experiments showing that IL-6 alone is not sufficient to affect food intake. Taken together, our findings provide novel information that helps us understand the effects of proinflammatory cytokines in neuronal populations responsible for the regulation of food intake and energy homeostasis.

## 4. Materials and Methods

### 4.1. Mice

The experiments performed were previously approved by the Ethics Committee on the Use of Animals of the Institute of Biomedical Sciences, University of Sao Paulo (protocol number: 73/2017; approved on 7 July 2017). To visualize AgRP neurons, AgRP-Cre mice (Stock No: 012899; The Jackson Laboratory, Bar Harbor, ME, USA) were bred with the Cre-inducible tdTomato-reporter mouse (Stock No: 007909, The Jackson Laboratory). To visualize POMC neurons, POMC-Cre mice (Stock No: 005965, The Jackson Laboratory) were bred with the Cre-inducible GFP-reporter mouse (Stock No: 004178, The Jackson Laboratory). AgRP-reporter and POMC-reporter mice were weaned at 3–4 weeks of age and genotyped via PCR using DNA extracted from the tail tip (REDExtract-N-Amp™ Tissue PCR Kit, Sigma, St. Louis, MO, USA). Only male mice were used in the experiments. Mice were produced and maintained in standard conditions (with 12-h light/dark cycles) and received filtered water and regular rodent chow *ad libitum*.

### 4.2. Brain Histology

To demonstrate the distribution of AgRP or POMC neurons in the ARH, adult mice were anesthetized with isoflurane and perfused transcardially with saline, which was followed by a 10% buffered formalin solution. Brains were collected and post-fixed in the same fixative for 60 min and cryoprotected overnight at 4 °C in 0.1 M PBS containing 20% sucrose. Brains were cut in 30-µm thick sections using a freezing microtome. In the histological experiment, a fluorescence reaction was performed to amplify GFP staining, as previously described [29]. No reactions were required to visualize the tdTomato reporter protein. Brain sections were mounted onto gelatin-coated slides and covered with Fluoromount G mounting medium (Electron Microscopic Sciences, Hatfield, PA, USA). Photomicrographs were acquired with a Zeiss Axioimager A1 microscope (Zeiss, Munich, Germany).

### 4.3. Electrophysiology

To examine the acute effects induced by TNF-α, IL-6, or IL-1β on the membrane excitability of AgRP and POMC neurons, whole-cell patch-clamp recordings were performed in hypothalamic slices of adult (8-week-old to 12-week-old) AgRP-reporter and POMC-reporter male mice. Animals were decapitated, their brains were collected, and immediately submerged in ice-cold, carbogen-saturated (95% O_2_ and 5% CO_2_) artificial cerebrospinal fluid (aCSF, 124 mM NaCl, 2.8 mM KCl, 26 mM NaHCO_3_, 1.25 mM NaH_2_PO_4_, 1.2 mM MgSO_4_, 5 mM glucose, and 2.5 mM CaCl_2_). Coronal sections (250-µM thick) from a hypothalamic block were cut with a vibratome (Leica Biosystems, model: VT1000S, Buffalo Grove, IL, USA; RRID: SCR_016495) and then incubated in oxygenated aCSF at room temperature for at least 1 h before recording. Slices were transferred to the recording chamber and allowed to equilibrate for 10–20 min before the recording. The slices were bathed in oxygenated aCSF (30 °C) at a flow rate of 2 mL/min. In the current-clamp mode, neurons were recorded under zero current injection (I = 0). The pipette solution contained 120 mM K-gluconate, 1 mM NaCl, 10 mM KCl, 10 mM HEPES, 5 mM EGTA, 1 mM CaCl_2_, 1 mM MgCl_2_, 3 mM KOH, and 4 mM (Mg)-ATP, pH 7.3. The input resistance was assessed by measuring the voltage deflection at the end of the response to a hyperpolarizing rectangular current pulse (500 ms of −10 to −30pA). The resting membrane potential was monitored for at least 5 min (Basal, Mexico), followed by the addition of mouse recombinant TNF-α (20 ng/mL, Sigma-Aldrich, St. Louis, MO, USA), mouse recombinant IL-6 (10 ng/mL, Sigma-Aldrich), or mouse recombinant IL-1β (17.5 ng/mL, Abcam, Cambridge, UK) to the bath for approximately 5 min. The effects of TNF-α, IL-6, and IL-1β on RMP were monitored for up to 15 min (washout period). Additionally, we determined whether the co-administration of IL-6 produces changes in the responses induced by IL-1β on AgRP neurons. The membrane potential values were compensated to account for the junction potential (−8 mV). Only one cell was recorded in each brain slice.

### 4.4. Data Analysis

GraphPad Prism software (GraphPad, San Diego, CA, USA; RRID: SCR_002798) was used for statistical analysis. The effects of the cytokines were evaluated by paired two-tailed Student’s *t*-test. Results were expressed as a mean ± standard error of the mean.

## Figures and Tables

**Figure 1 ijms-21-08928-f001:**
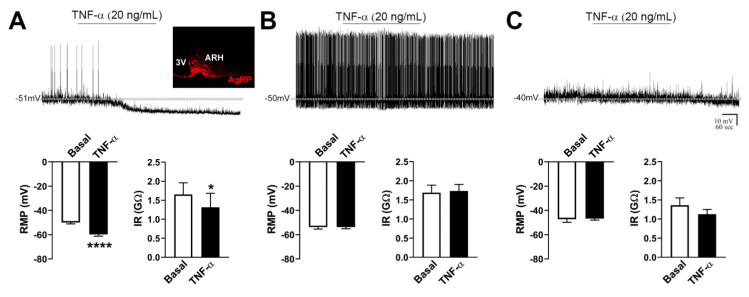
Tumor necrosis factor α (TNF-α) acutely inhibits the activity of agouti-related peptide (AgRP)-producing neurons. (**A**) Acute effects of TNF-α on the resting membrane potential (RMP) and input resistance (IR) of responsive AgRP neurons (*n* = 6). In inset shows AgRP-producing neurons in the ventromedial part of the arcuate nucleus (ARH). 3V = third ventricle. (**B**) Absence of changes in RMP and IR in nonresponsive AgRP neurons that were active (firing action potentials) during baseline (*n* = 11). (**C**) Absence of changes in RMP and IR in AgRP neurons that were quiescent during baseline (*n* = 4). In all conditions, representative whole-cell patch-clamp recordings are shown. Dashed lines indicate baseline RMP. TNF-α was applied to the bath for approximately 5 min. * *p* = 0.0391, **** *p* < 0.0001.

**Figure 2 ijms-21-08928-f002:**
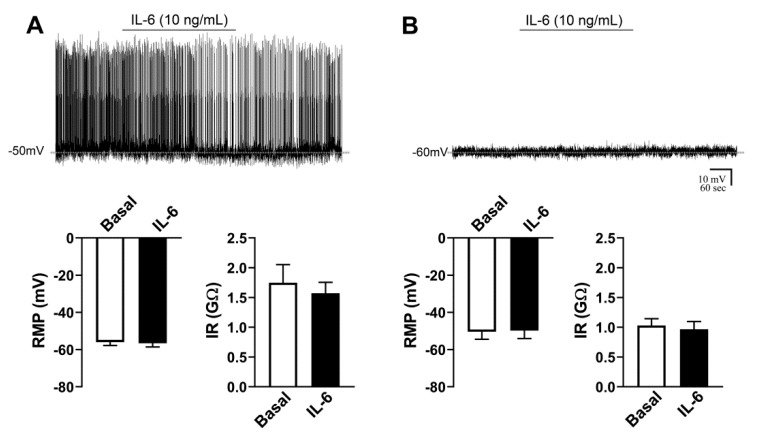
Activity of AgRP neurons is not acutely affected by interleukin-6 (IL-6). (**A**,**B**). IL-6 does not induce acute changes in the resting membrane potential (RMP) or input resistance (IR) of active (firing action potentials during baseline, *n* = 5) and quiescent (B, *n* = 5) AgRP-expressing neurons. Dashed lines indicate baseline RMP. IL-6 was applied to the bath for approximately 5 min.

**Figure 3 ijms-21-08928-f003:**
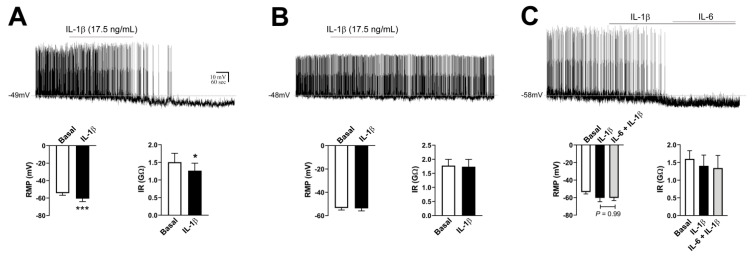
Interleukin-1β (IL-1β) inhibits the activity of AgRP neurons. (**A**). Acute effects of IL-1β on the resting membrane potential (RMP) and input resistance (IR) of responsive AgRP neurons (*n* = 8). (**B**). RMP and IR of nonresponsive AgRP neurons (*n* = 11). (**C**). Combined effect of IL-6 and IL-1β on the RMP and IR of responsive AgRP neurons (*n* = 3). In this last case, brain slices were initially treated with IL-1β, followed by application of both IL-6 and IL-1β. In all conditions, representative whole-cell patch-clamp recordings are shown. Dashed lines indicate baseline RMP. IL-1β or IL-6 + IL-1β were applied to the bath for approximately 5 min. * *p* < 0.05, *** *p* = 0.0007.

**Figure 4 ijms-21-08928-f004:**
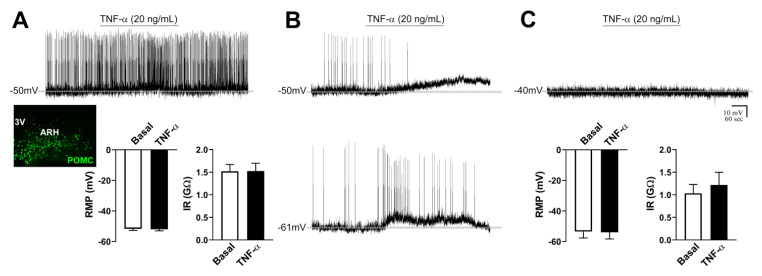
TNF-α causes mild effects on the activity of proopiomelanocortin (POMC) neurons in the arcuate nucleus. (**A**). Absence of changes in the resting membrane potential (RMP) and input resistance (IR) of nonresponsive POMC neurons that were active (firing action potentials) during baseline (*n* = 15). In inset shows POMC-expressing neurons in the lateral part of the arcuate nucleus (ARH). 3V = third ventricle. (**B**). Representative whole-cell patch-clamp recordings of two POMC neurons from the same animal that depolarized after TNF-α application. This effect was reversible during the washout period. (**C**). Absence of changes in RMP and IR in quiescent POMC neurons (*n* = 4). Dashed lines indicate baseline RMP. TNF-α was applied to the bath for approximately 5 min.

**Figure 5 ijms-21-08928-f005:**
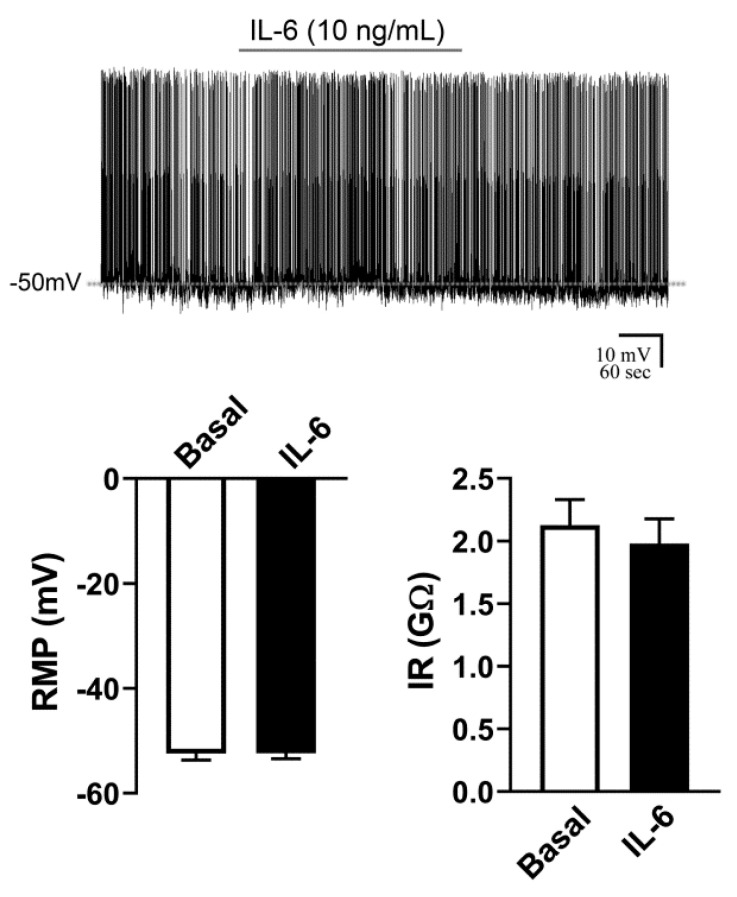
IL-6 causes no acute changes in the resting membrane potential (RMP) and input resistance (IR) of POMC neurons (*n* = 14) in the arcuate nucleus. All recorded POMC neurons were active at baseline. Dashed line indicates baseline RMP. IL-6 was applied to the bath for approximately 5 min.

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
