# Peer review of "Tumor Necrosis Factor α and Interleukin-1β Acutely Inhibit AgRP Neurons in the Arcuate Nucleus of the Hypothalamus"

_ijms, 2020, doi:10.3390/ijms21238928_

Round 1
Reviewer 1 Report
The manuscript from Chaves and coauthors has been significantly improved. However, due to adding some new information the proper experiments and corrections should be done.
- When adding IL-1 beta data the authors should be consistent and check its effect also on acitivity of POMC neurons. One can expect it from reading the summary and conclusions.
- The authors argue against synergistic effect of IL1beta and IL6. But they show a representative image (Figure 3c), which doesn't support this idea. It looks like that these ILs do act synergistically. I expect that the IL-1beta effect did not have enough time to develop. Please change the recording image to the porper one, where one can see comparable effects for applicatons of IL-1beta alone and IL-6 together with IL-1beta.
Overall, I find that these corrections can be done relatively fast and I am now very positive to see a resubmitted version soon.
Author Response
Response to point #1: We would like to thank the reviewer for his/her suggestions. IL-1β effects on the activity of POMC neurons were investigated by Scarlett et al. (Endocrinology 148:4217-4225, 2007). These authors showed that intracerebroventricular infusion of IL-1β induced Fos expression in ARH POMC neurons. Additionally, electrophysiological recordings demonstrated that IL-1β increased the frequency of action potentials and the release of α-MSH of ARH POMC neurons. Our observation that IL-1β causes an inhibitory effect in AgRP/NPY neurons is consistent with the well-known antagonistic role of POMC and AgRP/NPY neurons on feeding (Ollmann et al., Science 278:135-138, 1997). Since the role of IL-1β on POMC neurons has already been studied, we added a paragraph in the Discussion comparing our results on AgRP neurons with previous data studying IL-1β effects on POMC neurons (Scarlett et al., Endocrinology 148:4217-4225, 2007).
Response to point #2: We apologize for showing a confusing representative image. We replaced the image (Figure 3C) to another that represents better the lack of synergistic effect between IL-1β and IL-6.
Reviewer 2 Report
None
Author Response
RESPONSE: We thank the reviewer for their initial comments.
Round 2
Reviewer 1 Report
The manuscript can be accepted in the present form.
This manuscript is a resubmission of an earlier submission. The following is a list of the peer review reports and author responses from that submission.
Round 1
Reviewer 1 Report
Dear Authors,
My major problem with your study is that I can not see any novelty: similar electrophysiological results (and with better explanation) were previously published for AGRP neurons (Hao L et al, Brain Res, 2016) and for POMC neurons (Yi CX et al, Nat Communications, 2017). As for Figure 1 - it is so basic thing, that I doubt that it can be presented as a separate figure. Altogether, I can not support publication in the current form.
Reviewer 2 Report
The paper by Chaves et al entitled ‘Tumor necrosis factor alpha acutely inhibits AgRP neurons in the arcuate nucleus of the hypothalamus’ identified that the pro-inflammatory cytokines regulate the activities of AgRP and POMC neurons. The main observation is that TNF-alpha induces an acute inhibition in the electrical activity of AgRP neurons, but not POMC neurons. In addition, authors observed that IL-6 does not affect the activities of Agrp and POMC neurons. Authors finally suggested that anorexia induced by acute systemic inflammation is associated with the firing of AgRP neurons, not POMC neurons.
Major comments:
- Authors described that “TNF-α may also enhance satiety by increasing the activity of a small group of POMC neurons ”However, it is not reasonable to include the context with the data showing that only two out of fifteen POMC neurons responded to TNF-α. In addition, a previous literature have shown that TNF-α significantly increased neuronal firing and evoked excitability of arcuate nucleus POMC neurons. Therefore, authors have to rephrase the statement or find out other clue to interpret the data.
- It has been well defined that inflammatory stimuli such as overnutrion (High fat diet, High calorie diet), saturated free fatty acids, lipopolysaccharide, inflammatory cytokines trigger the inflammatory responses in the hypothalamic neurons that control feeding and energy expenditure. Manuscript retains a scientific novelty showing electrical properties of Agrp neurons in response to major proinflammatory cytokines. It is informative to better understand the actions of the hypothalamic neuronal circuit during the development of acute inflammation. However, manuscript still has a lack of scientific novelty. In the discussion section (line 286- 300), authors discussed about the discrepancy between effects of IL-6 in this article and its anorexigenic impacts in other reporters and suggested further study exploring the effects of IL-6 an IL-1 beta combination on the neuronal activities of Agrp and Pomc neurons. Manuscript will be strengthen, if authors include some data regarding combination effects
- From the analysis of input resistance (IR), authors claimed that the alteration of agrp neuronal activity might be coupled to opening of ion channels such as K+and Cl- channels. However, it seems like over-interpretation without any experimental evidence. Thus, further electrophysiological experiments with the ion channel blockers are required for the statement
Minor comment:
Authors described that results were expressed as mean ± standard error of the mean (SEM) in statistical analysis of the method section. However, error bars of figure 2A (data for input resistance) do not reflect the statistical significance. Please double check whether the graph was created from the raw data performed with standard deviation (SD)